# Cryoballoon Ablation for Persistent and Paroxysmal Atrial Fibrillation: Procedural Differences and Results from the Spanish Registry (RECABA)

**DOI:** 10.3390/jcm11051166

**Published:** 2022-02-22

**Authors:** Ermengol Vallès, Jesús Jiménez, Julio Martí-Almor, Jorge Toquero, José Miguel Ormaetxe, Alberto Barrera, Arcadio García-Alberola, José Manuel Rubio, Pablo Moriña, Carlos Grande, Maria Fé Arcocha, Rafael Peinado, Rocío Cózar, Julio Hernández, Luisa Pérez-Alvarez, Larraitz Gaztañaga, Angel Ferrero-De Loma-Osorio, Ricardo Ruiz-Granell, Roger Villuendas, Jesús Daniel Martínez-Alday

**Affiliations:** 1Hospital del Mar, Universitat Autònoma de Barcelona, IMIM, 08003 Barcelona, Spain; jejilo78@gmail.com (J.J.); jmartialmor@gmail.com (J.M.-A.); 2Hospital Universitario Puerta de Hierro, 28222 Majadahonda, Spain; jorgetoqueroramos@secardiologia.es; 3Hospital Universitario de Basurto, 48013 Bilbao, Spain; josemiguel.ormaetxemerodio@osakidetza.net (J.M.O.); larraitz.gaztanaga@gmail.com (L.G.); txules@gmail.com (J.D.M.-A.); 4Hospital Universitario Virgen de la Victoria, 29010 Malaga, Spain; albarrera62@gmail.com; 5Hospital Clínico Universitario Virgen de la Arrixaca, 30120 Murcia, Spain; arcadi@secardiologia.es; 6Hospital Universitario Fundación Jiménez Díaz, 28040 Madrid, Spain; jmrubio@fjd.es; 7Hospital Juan Ramón Jiménez, 21005 Huelva, Spain; pmorina@gmail.com; 8Hospital Universitari Son Espases, 07010 Palma de Mallorca, Spain; carlose.grande@ssib.es; 9Hospital Universitario de Araba, 01009 Alava, Spain; mrf.arcocha@gmail.com; 10Hospital Universitario La Paz, 28046 Madrid, Spain; rpeinado@secardiologia.es; 11Hospital Universitario Virgen Macarena, 41009 Sevilla, Spain; rociocozarleon@hotmail.com; 12Hospital Universitario Nuestra Señora de la Candelaria, 38010 Santa Cruz de Tenerife, Spain; calula@telefonica.net; 13Complejo Hospitalario Universitario A Coruña, 15006 A Coruña, Spain; luisa.perez.alvarez@sergas.es; 14Clínica IMQ Zorrotzaurre, 48014 Bilbao, Spain; 15Hospital Clínico Universitario de Valencia, 46010 Valencia, Spain; angelferrero@hotmail.com (A.F.-D.L.-O.); ricardo.ruizgranell@gmail.com (R.R.-G.); 16Hospital Germans Trias i Pujol, 08916 Badalona, Spain; rogervilluendas@hotmail.com

**Keywords:** persistent atrial fibrillation, cryoballoon procedure, registry

## Abstract

Introduction: Cryoballoon ablation (CBA) has become a standard treatment for paroxysmal atrial fibrillation (PaAF) but limited data is available for outcomes in patients with persistent atrial fibrillation (PeAF). Methods: We analyzed the first 944 patients included in the Spanish Prospective Multi-center Observation Post-market Registry to compare characteristics and outcomes of patients undergoing CBA for PeAF versus PaAF. Results: A total of 944 patients (57.8 ± 10.4 years; 70.1% male) with AF (27.9% persistent) were prospectively included from 25 centers. PeAF patients were more likely to have structural heart disease (67.7 vs. 11.4%; *p* < 0.001) and left atrium dilation (72.6 vs. 43.3%; *p* < 0.001). CBA of PeAF was less likely to be performed under general anesthesia (10.7 vs. 22.2%; *p* < 0.001), with an arterial line (32.2 vs. 44.6%; *p* < 0.001) and assisted transeptal puncture (11.9 vs. 17.9%; *p* = 0.025). During an application, PeAF patients had a longer time to −30 °C (35.91 ± 14.20 vs. 34.93 ± 12.87 s; *p* = 0.021) and a colder balloon nadir temperature during vein isolation (−35.04 ± 9.58 vs. −33.61 ± 10.32 °C; *p* = 0.004), but received fewer bonus freeze applications (30.7 vs. 41.1%; *p* < 0.001). There were no differences in acute pulmonary vein isolation and procedure-related complications. Overall, 76.7% of patients were free from AF recurrences at 15-month follow-up (78.9% in PaAF vs. 70.9% in PeAF; *p* = 0.09). Conclusions: Patients with PeAF have a more diseased substrate, and CBA procedures performed in such patients were more simplified, although longer/colder freeze applications were often applied. The acute efficacy/safety profile of CBA was similar between PaAF and PeAF patients, but long-term results were better in PaAF patients.

## 1. Introduction

Pulmonary veins (PV) isolation has emerged as the cornerstone of atrial fibrillation (AF) ablation procedures [1]. Cryoballoon ablation (CBA) has shown comparable results to those obtained by radiofrequency ablation in patients with paroxysmal atrial fibrillation (PaAF) [2,3,4,5,6,7,8]. Since patients with persistent atrial fibrillation (PeAF) have a more complex cardiac substrate, multiple ablation strategies involving additional ablation beyond PV isolation have been described. However, stand-alone PV isolation has been shown to be non-inferior to more extensive ablation in this type of patient [9]. This suggests a role for CBA in PeAF patients, but little is known about the characteristics for patient selection and acute procedural outcomes [10]. We aimed to assess and compare clinical characteristics, anatomical features, procedural differences, complications and outcomes of patients with PeAF and PaAF undergoing CBA.

## 2. Methods

### 2.1. Description of Registry

The Spanish Registry of Cryoballoon Ablation (RECABA) is a multicenter, prospective, observational post-market registry including 29 selected sites, sponsored by Medtronic Ibérica. In total, 27 sites enrolled patients during an inclusion period of 2 years and 3 months between September 2016 and January 2019, during which patients were followed according to routine clinical practice.

Ethics Committee approval was obtained according to local legislation. The study was conducted in compliance with the most recent version of the Declaration of Helsinki, Spanish laws and regulations (Royal Decree 1090/2015, Royal Decree 1616/2009, Order SAS/3470/2009 of 16 December). This observational study did not require authorization by the Spanish Agency of Medicines and Medical Devices (AEMPS), as stipulated in Royal Decrees 1090/2015 and 1616/2009, since it is a clinical investigation with CE marked medical devices used in accordance with the clinical purpose of the device (Arctic Front Advance, Medtronic Inc.). The study was assessed and approved by the IRB, Comité Ético de Investigación Clínica de Euskadi (CEIC-E) on 9 May 2016, and by the Ethical Committee of Hospital de Mar, Comité Ético de Investigación Clínica del Consorci Mar Parc de Salut de Barcelona (CEIC-Parc De Salut Mar) on 21 June 2016. All patients signed informed consent before inclusion in the registry.

The aim of the study was to monitor the current use and outcomes of PV ablation with CBA procedures and collect real-world data on CBA procedures in patients with either PaAF or PeAF. Sites with demonstrable experience of the CBA technique (at least 10 procedures a year) were selected [11,12]. Clinical data were collected at the baseline procedure and at annual follow up through a web-based platform from the hospital patient files. Periodic data cleaning was performed to ensure data quality. The primary objective was to evaluate the efficacy of the CBA at 12 months, defined as the absence of clinical recurrences of AF documented in a 12-lead ECG or subclinical recurrences of AF documented by means of Holter monitoring lasting at least 30 s. Only AF was considered as an arrhythmia recurrence. Secondary objectives included evaluation of acute procedural endpoints, procedural complications, and utilization of healthcare resources.

### 2.2. Analysis Endpoints

This analysis was performed on the 944 patients (from the first 25 sites) enrolled during the first year of the study. There were no exclusion criteria. Patients were followed according to each center’s discretion (which most of the times included 3- and 6-month visits with ECG and 24-h Holter monitoring), and a 12 month follow-up visit was protocol required. The primary objectives of this subanalysis were to compare clinical characteristics, anatomical features, and procedural differences of patients with PeAF and PaAF undergoing CBA procedures. Acute success and 12-month freedom from a ≥30 s recurrence of documented AF after a 90-day blanking period were evaluated. All procedure-related complications over 12 month follow-up were recorded. Early and late complications, including left atrial flutter, were treated as required according to operator discretion. Major adverse cardiac events (MACE) were defined as the following events: acute myocardial infarction, ischemic stroke, cardiac tamponade, atrio-esophageal fistula and death.

### 2.3. Cryoballoon Ablation Procedure

Common elements of the CBA procedure have been previously described. In brief, the procedure was performed with the patient under sedation or total anesthesia and under infusion of unfractionated heparin guided by activated clotting time (ACT). A single trans-septal puncture was performed using a long sheath, guided by fluoroscopy and/or transesophageal/intracardiac echocardiography. The transeptal sheath was exchanged over a guidewire for a 15F deflectable introducer, and the second-generation cryoballoon (Arctic Front Advance; either the 23- or 28-mm diameter balloon) was introduced together with the inner-lumen circular mapping catheter (Achieve, Medtronic) into the antrum of each PV. Most of the procedures were performed under the guidance of a three-dimensional reconstruction of the left atrium and pulmonary veins, extracted from a pre-procedural cardiac MRI, CT scan, or left atrial angiography. All procedures aimed to achieve both PV entrance and exit block, which was assessed by careful manipulation and stimulation from the Achieve catheter into each pulmonary vein. Patients in AF during the procedure were cardioverted before or after PV isolation depending on operator preference. Regardless of the type of AF, PV isolation alone was performed. No adjunctive ablation was completed other than CTI ablation in case of history of typical flutter. In the rare cases of documentation of atrial arrhythmias other than AF during the CBA procedure the decision to proceed with the AF ablation vs. perform an electrical or farmacological cardioversion prior to AF ablation was taken at operator discretion.

### 2.4. Statistical Analyses

Sociodemographic and clinical data were gathered at the patient level, procedural and freeze application data were gathered at the intervention level. Consequently, each cryoapplication was considered individually, and when appropriate, cryoapplication data were aggregated by location or by patient and location. Frequencies and percentages were used to describe categorical variables. Means, medians, standard deviation and interquartile ranges were used to describe continuous variables. For categorical variables, groups of patients or categories were compared using chi-square tests and standardized adjusted residuals were reported to assess deviations from marginal expected frequencies. For continuous variables, t-tests, and ANOVA procedures were used to compare groups, while multiple comparisons were carried one using Bonferroni adjustment. Variance equality was tested using Levene test. When needed, non-parametric tests were used to compare medians. No imputation method was used and missing information was considered lost pair-wise by the combination of variables considered. When comparing prevalence figures (such as complications) observed values were tested against the overall proportional distribution of cases in the comparison groups. An α = 0.05 nominal significance level was considered for all tests.

## 3. Results

### 3.1. Population Characteristics

We analyzed the first 944 patients included in the RECABA registry. Table 1 shows the clinical and anatomical characteristics of the patient population. Of note 70.1% of patients were male and mean age was 57.8 ± 10.4 years. Only 17.2% had structural heart disease (SHD). Overall, 27.9% procedures were performed for PeAF and 5.7% for repeat ablation patients. Antiarrhythmic drugs (AAD) and anticoagulation were used in 80.3% and 73.7% of patients, respectively. Left atrium dilation (area > 20 cm^2^) was seen in 48.6% of patients and 15.9% had a left common PV ostium.

### 3.2. Procedural Characteristics

All patients in the 29 participating centers received a CBA procedure. Table 2 describes the procedural characteristics. Of note, general anesthesia and adjunctive imaging during transeptal puncture (using intracardiac or transesophageal echocardiography) were used only in 18.9% and 16.3% of patients, respectively. Phrenic nerve function was monitored in 100% of cases using diaphragmatic pacing. Pulmonary vein potentials were visualized in 59.8% of PVs during the ablation. The number of freeze applications per vein was 1.84 ± 1.07, mean time to effect (TTE) was 54.4 ± 37.2 s, and mean minimal temperature (minT) of the balloon was −48.95 ± 6.6 °C. Overall, 97.85% of PVs were acutely isolated. Complications are listed in Table 3, separating adverse events as procedure-related (5.08%), non-procedure-related (2.64%), and MACE (acute myocardial infarction, ischemic stroke, cardiac tamponade, atrio-esophageal fistula and death; 0.3%). Complications observed during the procedure consisted of 16 patients with transient phrenic nerve injury (resolved by the time of procedure discharge), 10 patients with transient ST segment elevation, 5 patients with phrenic nerve injury unresolved at the time of procedure discharge, 2 patients with pericardial effusion, 1 groin hematoma, 1 hemorrhage requiring transfusion, 1 ventricular tachycardia and 1 femoral vein laceration. Complications observed early after the procedure consisted of 5 patients with groin hematoma, 1 arterial embolism, 1 myocardial infarct, 1 gastric complication and 1 arteriovenous fistula. Patients were discharged at a mean 1.59 ± 9.46 days, 65% on AAD, with no differences between PaFA vs. PeFA (69.6% vs. 63.3%; *p* = 0.072) and 100% on anticoagulation.

### 3.3. Clinical and Anatomical Differences between PaAF and PeAF Patients

Variables were compared between groups, and the following were statistically significant (Table 1): PeAF was more frequent among males (80.1 vs. 66.7%, *p* < 0.001), and associated with hypertension (53.8 vs. 44.2%, *p* = 0.008), hypercholesterolemia (41.9 vs. 32.2%, *p* = 0.005), structural heart disease (67.7 vs. 11.4%, *p* < 0.001), previous pacemaker implant (6.2 vs. 2.7%, *p* = 0.012), alcohol use (28.7 vs. 20.9%, *p* = 0.003), tobacco use (16.1 vs. 10.6%, 0 = 0.024) and sleep apnea (17.8 vs. 10.4%, *p* = 0.003). PaAF was more frequent among patients with more than 5 years since AF diagnosis arrhythmia (29.68 vs. 19.84%, *p* < 0.001) and in participants who partake in high-intensity exercise with more than 300 min/week (46.9 vs. 32.8%, *p* = 0.002). Anticoagulants and beta-blockers were more commonly used prior to the CBA procedure in PeAF patients, (89.6 vs. 67.5% and 76.4 vs. 64.2%, respectively, *p* < 0.001 for both). AAD use was more common in PaAF (85.3 vs. 67.4%, *p* < 0.001), except for amiodarone, which was more often used in PeAF. PeAF was more associated with left ventricle (LV) dysfunction (27.6 vs. 3.7%, *p* < 0.001), LV hypertrophy (21.4 vs. 10.5%, *p* < 0.001) and LA dilation >20 cm^2^ (72.6 vs. 43.3%, *p* < 0.001). Interestingly there were no differences in the rate of anatomical variants such as left common ostium. 

### 3.4. Procedural Differences between PaAF vs. PeAF Patients

A number of differences were found between the two types of patients concerning the CBA procedure (Table 2). Patients with PaAF were more likely to undergo preprocedural imaging (70.2 vs. 56.5%, *p* < 0.001), to be treated under general anesthesia (22.2 vs. 10.7%, *p* < 0.001), to have an arterial line (44.6 vs. 32.2%, *p* < 0.001), and to undergo adjunctive monitoring during transseptal puncture (17.9 vs. 11.9%, *p* = 0.025). PaAF were also more likely to receive bonus CBA freeze applications (41.1 vs. 30.7%, *p* < 0.001), have a waiting time after isolation (29.5 vs. 20.2%, *p* = 0.005) and have adenosine testing performed (5.4 vs. 1.6%, *p* = 0.012). Patients with PeAF had a longer time to −30 °C during freezing (35.91 ± 14.20 vs. 34.93 ± 12.87 s, *p* = 0.021), and a colder balloon nadir temperature at vein isolation (−35.04 ± 9.58 vs. −33.61± 10.32 °C, *p* = 0.004). No differences were observed in the acute PV isolation rate (98.12 vs. 97.58% in PaAF vs. PeAF, respectively; *p* = 0.327) or procedural complications (Table 3). 

### 3.5. Long Term Results among PaAF and PeAF

In this case, 14 patients were lost during follow-up. Overall, 76.7% (217/930) of patients did not have a recurrence of AF over 15 month follow-up. PaAF patients had a higher percentage of freedom from AF recurrence than PeAF patients (78.9% vs. 70.9%, respectively; *p* = 009). Kaplan-Meier survival analyses showed significant curves separation (*p* = 0.015) corresponding to a faster rate of recurrence in PeAF patients, which was more accentuated after 10 months from the CBA procedure (Figure 1).

## 4. Discussion

The RECABA prospective registry of outcomes of cryoablation included a mostly young and healthy population, yet almost one third of patients had PeAF and almost half had some degree of LA enlargement. Most procedures were performed without general anesthesia and without adjunctive imaging during the transseptal puncture. Acute and follow-up success rates were high and comparable to those achieved with RF. Finally combined procedural complications, MACE, and late complication rates after cryoballoon ablation were low.

Few studies have analyzed CBA results and complications in patients with PeAF [13,14,15]. To our knowledge this is the only CBA registry comparing procedural differences and dosing parameters between PaAF and PeAF in the general population.

### 4.1. Comparison between PaAF and PeAF Patients

Persistent AF, as expected, was more frequently comorbid to other cardiovascular risk factors, such as hypertension, hypercholesterolemia, and tobacco/alcohol use. It was also associated with LV dysfunction, LV hypertrophy, LA dilation and previous pacemaker implantation. Surprisingly, although PeAF patients had more comorbidities and cardiovascular disease, PaAF was more frequent among patients with more than 5 years since AF diagnosis. This likely reflects selection bias within the registry and suggests operators prefer to be more aggressive in the interventional management of PaAF versus PeAF when the diagnosis has been made late in the evolution of the disease.

### 4.2. Procedural Differences between PaAF vs. PeAF Patients

A number of procedural parameters reinforce the theory that operators used a more involved approach in patients with PaAF, including: the performance of an advanced preprocedural image technique, the use of general anesthesia, arterial line, and/or advanced imaging during transseptal puncture. This was not only observed in pre-procedural planning techniques, but also during the ablation procedure itself. Patients with PaAF were more likely to have a wait time after isolation, to have an adenosine testing performed, and to receive bonus applications to the PVs. This is probably the main reason why no differences existed between PaAF and PeAF in terms of fluoroscopy time. It appeared that operators used more adjunctive methods to achieve durable isolation in PaAF patients, despite longer times to −30 °C and colder balloon nadir temperatures to isolate the veins in PeAF patients. The authors speculate longer times to −30 °C could be related to a larger pulmonary vein antrum size, resulting in poor balloon occlusion, limiting the cooling effect of the balloon application. Additionally, the fact that more 28-mm cryoballoons were used in patients with PeAF may indicate the intended strategy of creating a wider lesion set in this type of cardiac substrate. 

### 4.3. Complications and Acute Results. Comparison with Other Registries

Procedure-related complications have been reported in 5.5 to 9% of patients undergoing CBA procedures for PeAF [10,15,16,17], which is comparable to the complication rate in patients with PaAF. Phrenic nerve injury, mostly transient, has been the most common complication in CBA procedures, as frequent as 6.3% in initial reports, but has significantly decreased progressively over years. The German Ablation Registry of paroxysmal AF ablation showed an acute success rate similar for CBA and for radiofrequency, around 97.5% [5], also with similar complications rate, around 9.3%. Similarly, the 1STOP Italian registry study observed similar vein isolation and complications rates, but the latter included procedures performed with the first-generation cryoballoon [15]. None of these studies were performed to compare procedural characteristics of patients with PeAF versus PaAF. However, a subanalyses of the 1STOP [18] study included 486 patients with PeAF, and showed similar results to our study (8% rate of total complications).

### 4.4. Long-Term Results. Comparison with Other Studies

There is growing evidence regarding the importance of CBA as a firt line therapy for AF [19,20], but again the evidence is limited in patients with PeAF versus PaAF. CBA to achieve PVI for patients with PeAF has resulted in similar success rates as radiofrequency ablation with a single procedure, with approximately 55–60% of patients free of arrhythmia recurrences at 1 year follow-up [13,16]. Patients with <1 year of PeAF tend to have even greater AF recurrence-free rates [14]. Success rates with multiple procedures can reach 75% at 12 months [18] and 69% at 16 months [21], which is only slightly inferior to the success achieved in PaAF. Overall 76.7% of patients in our registry were free of AF recurrences at 15 months follow-up and patients with PaAF showed slightly better outcomes than patients with PeAF (78.9% vs. 70.9%, respectively). This seems reasonable since patients with PeAF had a more diseased substrate. Long-term success in PeAF patients was higher in our registry than have been previously reported. We believe this could be related to a younger population treated within RECABA (i.e., age of 57 ± 10 years in our registry vs. 65 ± 9 years in STOP Persistent AF Trial [15]).

### 4.5. Limitations

The limitations of this multicenter observational study include potential bias in patient selection, patient treatment, and the lack of a control group. Nevertheless, possible biases are mitigated by the prospective data collection and predefined data analysis plan. Since we aimed to describe the real-world results achieved in standard clinical practice, in this analysis a minimum procedure per year per center was requested so that the operator’s learning curve had been taken into account. All RECABA centers had already established experience with CBA before patient inclusion started; however, it is possible individual operators experiences a learning curve over the study period. The large number of patients included in the present analysis likely balance the possibility of a learning curve bias. Despite the use of cardiac magnetic resonance in several patients we did not perform a specific study of atrial fibrosis since this kind of study is not available in the majority of participating centers. Similarly measurements of the PV antrum were not systematically performed, therefore we cannot confirm that longer times to −30 °C are related to a larger pulmonary vein antrum size. Follow-up was at center’s discretion but the big majority of patients received at least 3 visits including 12-lead ECG and at least one 24h Holter monitor. Lastly this register did not include any patient performed with a zero X-ray ablation, even when this approach is increasing [22] and despite the non-negligible effects of X-ray on health [23,24].

## 5. Conclusions

Patients with PeAF undergoing CBA have more comorbidities and more diseased cardiac substrate. However, CBA procedures performed in patients with PeAF tend to be more simplified, although longer and/or colder freeze applications were frequently observed. The safety profile of CBA was similar for PaAF and PeAF. While acute efficacy is similar, long-term freedom from AF is higher in PaAF than in PeAF, but a high rate of freedom from AF was also observed in the PeAF cohort, with ≥70% free from AF at 15-month follow-up.

## Figures and Tables

**Figure 1 jcm-11-01166-f001:**
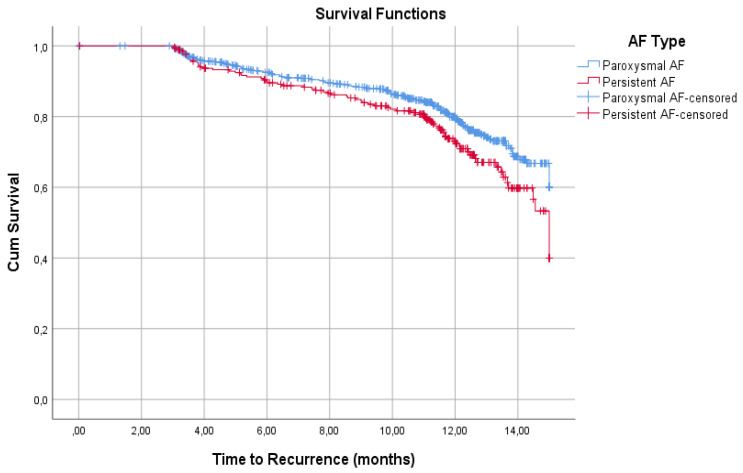
Recurrences Kaplan-Meier survival analyses. PaAF patients had a higher percentage of freedom from AF recurrence than PeAF patients at 15 months (78.9% vs. 70.9%, respectively). The curves show significant differences (*p* = 0.015), which are more accentuated after 10 months.

**Table 1 jcm-11-01166-t001:** Clinical and Anatomical Characteristics: Paroxysmal vs. Persistent.

Variable	Patients (*n* = 944)	PaAF (*n* = 681)	PeAF (*n* = 263)	*p* Value
Age (%)				
65–75 years	26.1%	26%	26.3%	0.48
>75 years	2.6%	2.5%	2.7%	0.52
Gender (male)	70.1%	66.7%	80.1%	<0.001
Time Since AF Diagnosis:				<0.001
<1 year	12.8%	10.8%	17.9%	
1–2 years	21.9%	22.6%	20.2%	
2–5 years	35.6%	35.7%	35.5%	
5–10 years	18.1%	19.6%	14.1%	
>10 years	8.8%	10%	5.7%	
Type of procedure (redo)	5.7%	6.2%	4.2%	0.23
Cardiac heart failure (HFpEF or HFrEF)	7.1%	2.8%	18.3%	<0.001
Hypertension	46.9%	44.2%	53.8%	0.008
Hypercholesterolemia	34.9%	32.2%	41.9%	0.005
Diabetes mellitus	9.1%	8.3%	10.3%	0.19
Vascular disease	5.6%	5.5%	5.7%	0.49
Prior stroke	5%	4.6%	6.1%	0.21
Presence of SHD	17.2%	11.4%	67.7%	<0.001
Any level of sport/exercise practice	42.9%	46.9%	32.8%	0.002
High level of alcohol use ^1^	23.1%	20.9%	28.7%	0.003
Any tobacco use	12.1%	10.6%	16.1%	0.02
Diagnosed OSAS	12.5%	10.4%	17.8%	0.003
Current AAD use:				<0.001
None	19.7%	14.7%	32.6%	
Any	80.3%	85.3%	67.4%	
Current AV Blockade drugs:				
Betablockers	67.6%	64.2%	76.4%	<0.001
Calcium-antagonists	6.2%	5.1%	9%	0.02
Current AC drugs	73.7%	67.5%	89.6%	<0.001
Echo LVEF:				<0.001
<35%	4.6%	1.1%	13.6%	
36–50%	5.8%	2.6%	14%	
>50%	89.6%	96.3%	72.3%	
Echo LVH	5%	10.5%	21.4%	<0.001
Echo LA dilatation (>20 cm^2^)	48.6%	43.3%	72.6%	<0.001
Dilated LA area:				<0.001
Mild (21–30 cm^2^)	66.1%	78%%	50%	
Moderate (31–40 cm^2^)	26.3%	17.2%	38.8%	
Severe (>40 cm^2^)	7.5%	4.8%	11.2%	
PV anatomy:				
Left common ostium	15.9%	16.4%	14.6%	0.43
More than 2 right PVs	8.1%	7.4%	8.8%	0.71

^1^ High level of alcohol use: for men, consuming more than 4 drinks on any day or more than 14 drinks per week; for women, consuming more than 3 drinks on any day or more than 7 drinks per week. AAD: antiarrhythmic drugs; AC: anticoagulant; AF: atrial fibrillation; AV: atrioventricular; HFpEF: heart failure with preserved ejection fraction; HFrEF: heart failure with reduced ejection fraction; LVEF: left ventricle ejection fraction; LA: left atrium; LVH: left ventricle hypertrophy; OSAS: obstructive sleep apnea syndrome; PaAF: paroxysmal atrial fibrillation; PeAF: persistent atrial fibrillation; PV: pulmonary vein; SDU: Standard Drink Unit; SHD: structural heart disease. 1High level of alcohol use >4 SDU/day (men), >2 SDU/day (women).

**Table 2 jcm-11-01166-t002:** Procedural Characteristics: Paroxysmal vs. Persistent.

	Patients (*n* = 944)	PaAF (*n* = 681)	PeAF (*n* = 263)	*p* Value
Advanced image technique	66.4%	70.2%	56.5%	<0.001
Anesthesia:				<0.001
General Anesthesia	18.9%	22.2%	10.7%	
Sedation	81.1%	77.8%	89.3%	
Arterial line	41.2%	44.6%	32.2%	<0.001
Number of catheters (excluding CB):				0.23
1	15.8%	17.4%	11.9%	
2	49.5%	48.2%	52.7%	
3	32.4%	32%	33.5%	
>3	2.3%	2.4%	1.9%	
Adjunctive imaging during transseptal puncture	16.3%	17.9%	11.9%	0.02
Type of assisted puncture:				0.26
TEE	59.9%	62.5%	51.6%	
ICE	40.1%	37.5%	48.4%	
Type of phrenic nerve monitoring:				0.05
Palpation	98.6%	98.3%	99.2%	
Fluoroscopy	27.7%	27.9%	21.8%	
Modified DI	23.7%	24.4%	21.8%	
Basal procedural rhythm:				<0.001
Sinus rhythm	76.7%	92.9%	35%	
Atrial fibrillation	22.5%	7.4%	61.5%	
Typical AFL	1.6%	0.9%	3.5%	
Atypical AFL	0.3%	0.5%	0%	
Total procedure time (min; ave ± SD)	117.8 ± 40.7	117.9 ± 40.3	117.8± 41.8	0.94
Dwell LA time (min; ave ± SD)	78.4 ± 28.1	78.3 ± 27.6	78.5 ± 29.3	0.90
Fluoroscopy time (min; ave ± SD)	25.8 ± 17.3	25.6 ± 16.3	26.9 ± 19.6	0.30
Application time (min; ave ± SD)	21.5 ± 8.2	21.2 ± 8.0	22.2 ± 8.7	0.07
Cryoballoon size:				0.001
28 mm	92.2%	90.5%	96.5%	
23 mm	10.6%	12%	7.3%	
Number of PV treated (ave ± SD)	3.98 ± 0.75	3.95 ± 0.76	4.04 ± 0.746	0.08
Applications per vein (ave ± SD)	1.87 ± 1.07	1.87 ± 1.02	1.86 ± 1.18	0.65
Time to −30 °C (sec; ave ± SD/median)	35.2 ± 13.3/32	34.9 ± 12.8/32	35.9 ± 14.2/33	0.02
PV potentials monitored:				
Total	59.8%	59.2%	61.2%	0.28
Left com. ostium	70.8%	74.7%	56.5%	0.09
LSPV	71.7%	74.4%	65.2%	0.009
LIPV	61.2%	59.2%	66.2%	0.06
RSPV	56.6%	55.7%	58.7%	0.41
RIPV	48%	45.7%	53.8%	0.03
Time to effect (sec; ave ± SD/median)	54.4 ± 37.2/44	53.8 ± 37.7/42	56.2 ± 35.8/47	0.18
Temperature at TTE (°C; ave ± SD/median)	−34 ± 10.1/−36	−33.6 ± 10.3/−35	−35 ± 9.5/−37	0.004
Pulmonary vein isolation % (ave ± SD)	97.8 ± 9.5	98.1 ± 8.9	97.5 ± 10.1	0.32
Application time per vein (sec; ave ± SD/median)	338.2 ± 182.9	335.7 ± 176.2/300	344.1 ± 199.1/263	0.24
Balloon MinT (°C): ave ± SD/median	−48.9 ± 6.6/−48	−48.8 ± 6.9/−48	−49.1 ± 6.8/−49	0.36
Balloon Rewarming time (sec): ave ± SD/median	39.7 ± 19.6/37	39.6 ± 19.1/37	39.7 ± 20.4/36	0.93
PV with bonus application	38.2%	41.1%	30.7%	<0.001
Waiting time after PV isolation	26.9%	29.5%	20.2%	0.005
Median waiting time (min)	15	15.3	16.5	0.34
Adjunctive CTI ablation	8.6%	8.7%	7.9%	0.67
Iodinated contrast (mL): ave ± SD	56.4 ± 46	56.8 ± 46.7	54.7 ± 44.2	0.57
Adenosine test	4.3%	5.4%	1.6%	0.01
Intraprocedural electrical cardioversions	28.7%	14.1%	65.6%	<0.001
Protamine use	52.9%	52.2%	54.7%	0.50
Z-Suture	72.7%	73.7%	70.2%	0.28

AFL: atrial flutter; CB: cryoballoon; CTI: cavo-tricuspid isthmus; ICE: intracardiac echography; LA: left atrium; MinT: Minimal temperature; PaAF: paroxysmal atrial fibrillation; PeAF: persistent atrial fibrillation; PV: pulmonary vein, Rewarming time: time from ablation off to balloon deflation; TEE: transesofageal echography; TTE: time to effect.

**Table 3 jcm-11-01166-t003:** Complications: Paroxysmal vs. Persistent.

	Patients (*n* = 944)	PaAF (*n* = 681)	PeAF (*n* = 263)	*p* Value
Procedure-related complications (*n*;%)	48; 5.08%	31; 4.58%	17; 6.49%	0.26
During procedure	38; 4.02%	25; 3.69%	13; 4.96%	0.41
Phrenic nerve injury resolved by discharge	16; 1.69%	12; 1.77%	4; 1.53%	0.77
Transient ST segment elevation	10; 1.05%	6; 0.89%	4; 1.53%	0.40
Phrenic nerve injury unresolved by discharge	5; 0.52%	2; 0.30%	3; 1.15%	0.11
Early after procedure (<30 days)	10; 1.05%	6; 0.89%	4; 1.53%	0.40
Groin hematoma	5; 0.52%	4; 0.59%	1; 0.38%	0.68
Embolism	1; 0.1%			-
Myocardial infarct	1; 0.1%			-
Gastric complication	1; 0.1%			-
Arteriovenous fistula	1; 0.1%			-
Non procedure-related complications (>30 days) (*n*;%)	25; 2.64%	14; 2.07%	11; 4.20%	0.07
Left atrial flutter	17; 1.80%	9; 1.33%	8; 3.05%	0.08
Other tachyarrhythmias	5; 0.52%	3; 0.44%	2; 0.76%	0.11
Major Adverse Cardiac Events (MACE) (*n*;%) (acute myocardial infarction, ischemic stroke, cardiac tamponade, atrio-esophageal fistula and death)	3; 0.31%	2; 0.30%	1; 0.38%	0.27
Total complications (*n*; %)	76; 8%	48; 7.09%	28; 10.69%	0.78

PaAF: paroxysmal atrial fibrillation; PeAF: persistent atrial fibrillation.

## Data Availability

The data that support the findings of this study are available from the corresponding author, E.V. with permission of the Medtronic team upon reasonable request.

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
