# Peer review of "Cryoballoon Ablation for Persistent and Paroxysmal Atrial Fibrillation: Procedural Differences and Results from the Spanish Registry (RECABA)"

_jcm, 2022, doi:10.3390/jcm11051166_

Round 1
Reviewer 1 Report
This manuscript by Valles et al. focused on the procedure differences and results of cryoballoon ablation (CAB) for patients with paroxysmal atrial fibrillation (PaAF) versus persistent AF (PeAF). Authors demonstrated that acute efficacy and safety profile of CBA were comparable between PaAF and PeAF, but long-term results on AF recurrence were better in patients with PaAF, using data from the Spanish Registry of Cryoballoon Ablation (RECABA). However, the difference in AF recurrence rate after CBA was small. As authors mentioned, although CBA has become standard treatment strategy for PaAF, its efficacy, safety, and long-term outcome for PeAF remains controversial. Therefore, the concept of this study is valuable and results seem agreeable. Although this study was well-conducted and manuscript seems written well, authors may want to resolve several issues as follows.
Major comments:
1) Although AF recurrence rate after CBA in patients with PeAF was significantly higher than that with PaAF, it seems reasonable because patients with PeAF had a more diseased substrate. Authors may want to comment about it in discussion.
Minor comments:
1) Abbreviation of CMR should be spelt out.
2) Although rates of free from AF recurrence in patients with PaAF and PeAF at 15 month were described as 78.9% and 70.9%, respectively, they look at 12 months on the Kaplan-Meier curve in Figure 1.
Author Response
Major comments:
1) Although AF recurrence rate after CBA in patients with PeAF was significantly higher than that with PaAF, it seems reasonable because patients with PeAF had a more diseased substrate. Authors may want to comment about it in discussion.
We agree with Reviewer 1 in the need for an explanation in terms of little differences in PeAF long-term results compared to PaAF. We have included it in the Discussion section (Page 9, paragraph 2, lines 11-12).
Minor comments:
1) Abbreviation of CMR should be spelt out.
We apologize for this mistake. In the reviewed manuscript we have changed CMR by cardiac magnetic resonance (page 9, paragraph 3, line 11).
2) Although rates of free from AF recurrence in patients with PaAF and PeAF at 15 month were described as 78.9% and 70.9%, respectively, they look at 12 months on the Kaplan-Meier curve in Figure 1.
We agree with the reviewer that the value for the proportion of patients free of AF after 15 months of follow up looks very similar to the proportion or surviving patients at month 12, but this fact is just a coincidence due to the small amount of AF events after 12 months of follow-up. For instance, in the case of Paroxysmal AF, the total number of cases free of AF recurrence after 15 months was P=0.789. That value is computed based on the total number of Paroxysmal AF patients (n=667) included in the study. On the other hand, the proportion of surviving patients is computed as a conditional probability, and it is based on the number of patients remaining under follow up at a certain moment. In the mentioned case, at 12 months of follow up, only n=288 Paroxysmal AF patients remain under observation. The rest of them had experienced recurrence or were lost from the follow up. It is a coincidence that the survival proportion of Paroxysmal patients at month 12 (P=0.794) is to be so close to the overall proportion of cases with no recurrence (P=0.789). Both probabilities are based in a different number of cases. In fact, you can see in the figure that the survival point at month 12 is just slightly below 0.80. Something similar happens with the Persistent AF group. The overall proportion free of recurrence after 15 months was 0.709, while the survival at month 12 was 0.727, slightly higher. If the reviewer wants us to include an explanation in the article or to report the survival at month 12, for readers to compare, we will be very glad to do so.
Reviewer 2 Report
I would congratulate with the authors for showing the results of RECABA prospective registry an outcomes of cryoablation. Results in 944 patients are extremely interesting and are consistent with the “real life” of persistent AF. I agree that procedures performed in such patients are more simplified, although longer/colder freeze applications are often applied. In this paper the acute efficacy/safety profile of cryo was similar between paroxysmal and persistent patients, but long-term results are better in paroxysmal. I have only few comments in order to improve the good manuscript
Methods: “Regardless of the type of AF, PV isolation alone was performed. No adjunctive ablation was completed other than CTI ablation in case of history of typical flutter”. However, authors should better clarify if a different LA arrhythmic substrate during ablation procedure has been documented, and the possible procedural approach (cardioversion? + drug therapy?). Please clarify this point, while authors well explained that complications (including left atrial flutter) were treated as required according to operator discretion
Results: It is extremely interesting how no statistically differences in fluoroscopy time between the two populations has been documented (25.6 ± 16.3 versus 26.9 ± 19.6). This concept should be also well explained in the discussion since today, for every EP physician, one of the most important limitation in cryoablation of AF persistent population may be a potential larger use of fluoroscopy, specially if compared to the zero X-ray ablation approach in electrophysiology in the same AF persistent population. Actually the existent “zero X-ray ablation approach” in electrophysiology as we definitely came in a new era (10.15420/aer.2020.02) should be considered. since the cryoablation, due to exposure related to X-ray transcatheter ablation carries small but non-negligible stochastic and deterministic effects on health (doi: 10.1093/europace/eux252 ; DOI: 10.1080/00015385.2020.1733303). Please cite this very important points, in limitations including all the suggested 3 references
Discussion: In the discussion I would add fundamental references about cryoablation as two Randomized Controlled trials from EARLY-AF Investigators (10.1056/NEJMoa2029980) and from Cryo-FIRST Investigators (10.1093/europace/euab029) showing the importance of cryo as first-line therapy, but for patients with paroxysmal atrial fibrillation. 2 suggested references should be updated as well
Author Response
Methods: “Regardless of the type of AF, PV isolation alone was performed. No adjunctive ablation was completed other than CTI ablation in case of history of typical flutter”. However, authors should better clarify if a different LA arrhythmic substrate during ablation procedure has been documented, and the possible procedural approach (cardioversion? + drug therapy?). Please clarify this point, while authors well explained that complications (including left atrial flutter) were treated as required according to operator discretion.
We thank the reviewer for this apreciation, which will improve the quality of the manuscript. We have now clarified the strategy in case of documentation of atrial arrhythmias other than AF in the Methods section (page 3, paragraph 2, lines 17-19).
Results: It is extremely interesting how no statistically differences in fluoroscopy time between the two populations has been documented (25.6 ± 16.3 versus 26.9 ± 19.6). This concept should be also well explained in the discussion since today, for every EP physician, one of the most important limitation in cryoablation of AF persistent population may be a potential larger use of fluoroscopy, specially if compared to the zero X-ray ablation approach in electrophysiology in the same AF persistent population. Actually the existent “zero X-ray ablation approach” in electrophysiology as we definitely came in a new era (10.15420/aer.2020.02) should be considered. since the cryoablation, due to exposure related to X-ray transcatheter ablation carries small but non-negligible stochastic and deterministic effects on health (doi: 10.1093/europace/eux252 ; DOI: 10.1080/00015385.2020.1733303). Please cite this very important points, in limitations including all the suggested 3 references
We agree with the reviewer in the need to clarify the reason why no differences were found in the fluoroscopy time between groups. This is now stated in the Discusion section (page 8, paragraph 4, lines 8-10).
We also agree with the lack of zero-x ray-exposure information in the previous version of the manuscript. We have added the required information and references in the Limitations (page 9, paragraph 3, lines 17-19) and in the References section (22 to 24).
Discussion: In the discussion I would add fundamental references about cryoablation as two Randomized Controlled trials from EARLY-AF Investigators (10.1056/NEJMoa2029980) and from Cryo-FIRST Investigators (10.1093/europace/euab029) showing the importance of cryo as first-line therapy.
We thank the reviewer for this appreciation. We have stated it in the Discusion section (page 9, paragraph 2, lines 2-3) and included these references in the References section (19,20).
Round 2
Reviewer 2 Report
Congratulations to the authors for the very good modified versions These concepts definitely improved the good manuscript